# Beef cattle that respond differently to fescue toxicosis have distinct gastrointestinal tract microbiota

Lucas R. Koester[1,2], Daniel H. Poole[3], Nick V. L. Serão[4]*, Stephan Schmitz-Esser[2,4]*

**1** Department of Veterinary Microbiology and Preventive Medicine, Iowa State University, Ames, IA, United States of America, **2** Interdepartmental Microbiology Graduate Program, Iowa State University, Ames, IA, United States of America, **3** Department of Animal Science, North Carolina State University, Raleigh, NC, United States of America, **4** Department of Animal Science, Iowa State University, Ames, IA, United States of America

* serao@iastate.edu (NVS); sse@iastate.edu (SSE)

**Data Availability Statement:** The 16S rRNA gene and ITS1 region sequences have been submitted to the NCBI Sequence Read Archive SRA and are available under the BioProject ID PRJNA498290.

## Abstract

Tall fescue (*Lolium arundinaceum*) is a widely used forage grass which shares a symbiosis with the endophytic fungus *Epichloë coenophiala*. The endophyte produces an alkaloid toxin that provides herbivory, heat and drought resistance to the grass, but can cause fescue toxicosis in grazing livestock. Fescue toxicosis can lead to reduced weight gain and milk yields resulting in significant losses to the livestock industry. The objective of this study was to identify bacterial and fungal communities associated with fescue toxicosis tolerance. In this trial, 149 Angus cows across two farms were continuously exposed to toxic, endophyte-infected, fescue for a total of 13 weeks. Of those 149 cows, 40 were classified into either high (HT) or low (LT) tolerance groups according to their growth performance (weight gain). 20 HT and 20 LT cattle balanced by farm were selected for amplicon sequencing to compare the fecal microbiota of the two tolerance groups. This study reveals significantly (*q<0.05*) different bacterial and fungal microbiota between HT and LT cattle, and indicates that fungal phylotypes may be important for an animal's response to fescue toxicosis: We found that fungal phylotypes affiliating to the Neocallimastigaceae, which are known to be important fiber-degrading fungi, were consistently more abundant in the HT cattle. Whereas fungal phylotypes related to the genus *Thelebolus* were more abundant in the LT cattle. This study also found more pronounced shifts in the microbiota in animals receiving higher amounts of the toxin. We identified fungal phylotypes which were consistently more abundant either in HT or LT cattle and may thus be associated with the respective animal's response to fescue toxicosis. Our results thus suggest that some fungal phylotypes might be involved in mitigating fescue toxicosis.

## Introduction

Tall fescue (*Lolium arundinaceum*) is a common cool season grass used widely as forage for grazing livestock in the southeastern United States. The grass shares a symbiosis with *Epichloë*

**Funding:** This work was supported by North Carolina Cattlemen's association and North Carolina Agricultural foundation. The funders had no role in study design, data collection and analysis, decision to publish, or preparation of the manuscript.

**Competing interests:** The authors have declared that no competing interests exist.

*coenophiala*, a fungus that grows as an endophyte within the plant and provides heat and herbivory resistance from both insects and mammals. The fungus produces ergot alkaloid compounds that have been shown to cause fescue toxicosis (FT) in grazing livestock species such as cattle, sheep and goats [1–3]. Many biologically active alkaloids are produced by the fungus, but ergovaline, an ergopeptide, is commonly thought to be responsible for FT [2]. However, until now, there is no clear evidence that ergovaline is the most or only responsible ergot alkaloid inducing FT–other ergot alkaloids may also contribute to FT. Induced by the consumption of these toxic ergot alkaloids, FT is a metabolic disease that, in ruminant livestock, manifests as vasoconstriction, higher body temperature, suppressed appetite, and reduced heart rate and prolactin levels [1, 3, 4]. This disease causes an estimated loss of $2 billion US dollars each year due to reduced body weight (BW), milk yields, and rate of calving [5].

Efforts to reduce or eliminate FT included removal of endophyte infected fescue and planting cultivars of endophyte-free [3] fescue. The endophyte free fescue improved cattle performance, but resulted in a general weakening of the plant, reducing tolerance to insects, nematodes, high temperatures and overgrazing [3]. Additionally, researchers focused on the endophyte, identifying strains that produce lower levels of the ergot alkaloids while still providing drought and insect resistance for the grass [3, 6, 7]. These studies showed that cattle and sheep fed fescue infected with low or non-ergot producing endophyte strains improved performance similar to endophyte-free fescue. These strains of fescue are being sold commercially [3, 6, 7]. Finally, management practices such as interseeding higher levels of different plants, rotational grazing, fertilizing with low nitrogen fertilizers and reducing seed heads that contain high-ergot producing endophyte strains in pasture were shown to reduce the toxin levels within the forage [3].

Another strategy to limit the negative effects of the tall fescue endophyte on cattle could be the identification of animals with greater tolerance to FT. Studies on host response of FT are still limited in the literature, with most of them focusing on breed differences [8–11]. Recently, using the same animals as in this study, Khanal et al. identified pregnant Angus cows showing distinct growth potential under FT [12]. Cows classified as tolerant to FT had higher growth and body condition score, and lower rectal temperature and hair coat score than susceptible animals. Using the same data, Khanal et al. identified 550 differentially expressed (DE) genes between tolerant and susceptible cows, with in which the most DE genes had functions such as regulation of vasoconstriction and hair coat shedding [13]. With respect to host genomics, the few reports available in the literature have focused on candidate gene approaches [14–16].

Studies published on the effect of toxic tall fescue effects on gastrointestinal (GI) microbiota are still limited. Most recently, Mote et al. surveyed fecal bacterial communities of beef cattle during a FT challenge, and described possible connections between the abundance of certain bacterial phylotypes and the host response to FT [17]. Other studies have suggested microbial communities within the cow rumen [18, 19], earthworm's intestine [20], and soil [21] are able to degrade ergovaline. It is thus conceivable that the GI tract microbiota may be able to reduce the toxic effect of ergot alkaloids and thus mitigate some of the impact of FT symptoms on livestock. We hypothesize that both the bacterial and fungal microbial communities within the GI tract are associated with FT tolerance.

In this study, we analyzed fecal microbial communities from cattle with contrasting growth performance during a chronic exposure FT challenge. Our goal was to identify shifts in bacterial, archaeal, and fungal microbial populations (using 16S rRNA gene and ITS1 region amplicon sequencing, respectively) between the two tolerance groups across two different locations.

## Materials and methods

### Ethics statement

All animal procedures were approved by the North Carolina State University (NCSU) Institutional Animal Care and Use Committee (protocol #13-093-A).

### Animal trial and selection of animals

An animal trial was conducted to gain insight on the effect of feeding toxic levels of endophyte infected forages. A total of 149 multiparous (parities 2 to 4) pregnant purebred Black Angus cows were used. Approximately half of the animals (78 cows) were located at the Upper Piedmont Research Station (UPRS–Reidsville, NC, NCSU), while the remaining animals (71 cows) were located at the Butner Beef Cattle Field Laboratory (BBCFL–Bahama, NC, NCSU).

Both groups had free access to forage and water during 13 weeks establishing a chronic exposure to toxic fescue (April to July 2016). Cattle at both locations grazed pastures known to be endophyte infected toxic tall fescue for the entirety of the study. Cattle were managed in a rotational grazing system and were moved to a new paddock every two weeks at each location to ensure adequate forage management as well as sufficient forage availability to the cows. Forage samples were collected every two weeks to evaluate nutrient quality and percentage of available forage that was fescue.

Fecal material was extracted from all cows following 13 weeks of exposure to endophyte-infected fescue. In brief, a lubricated shoulder length glove was inserted into the rectum and a grab sample of feces were collected from the colon. The fecal samples were labeled and placed immediately on ice and transported back to the lab for further processing. Fecal samples were transferred to labeled 15 ml polystyrene vials (BD Falcon) and stored at -80ºC for analysis.

Out of the 149 cows enrolled in the trial, 40 cows showing extreme growth performance were selected for further analyses. For each animal, growth during the trial was estimated as the slope of regression analysis of BW on weeks (average weekly gain; AWG). Slopes (i.e. AWG) were estimated based on 3 window periods: weeks 1 through 13 (w1_13), weeks 1 through 7 (w1_7), and weeks 7 through 13 (w7_13) to assess the effect of increase in temperature from April to July, availability of forage and exposure of infected tall fescue. The AWG data for each of these scenarios were analyzed using the following model:

$$AWG_{ijk} = \mu + L_i + P_j + b_1(iBW_k - \overline{iBW}) + e_{ijk} \qquad \text{[Eq 1]}$$

where $AWG_{ijk}$ is the AWG of the cow; $\mu$ is intercept; $L_i$ is the fixed effect of the $i$th location, $P_j$ is the fixed effect of the $j$th parity; $b_1$ is the partial regression coefficient for the covariate of initial BW ($iBW$); $iBW_k$ is the iBW of the kth cow; and $e_{ijk}$ is the residual associated with $y_{ijk}$, with $e_{ijk} \sim N(0, \sigma_e^2)$. Statistical analysis was performed in SAS 9.4 (Statistical Analysis System, Cary, NC, USA).

Identification of animals with high (HT) or low (LT) tolerance to FT were based on the residuals from Eq 1. The top (positive) 20 and bottom (negative) 20 residuals, with equal representation from each location (i.e. 20 from each location), were classified as HT and LT, respectively, for a total of 40 selected animals. Fecal samples from these 40 animals were subjected to amplicon sequencing targeting bacteria and fungi (see below). Then, the fecal microbiota of animals showing extreme performance (based on AWG) were compared, to achieve a clearer biological signal. Thus, a non-treatment group (which was not exposed to toxic fescue) was not included in this trial. Additionally, in our model to identify HT and LT we included the effects of parity and iBW of the cow in order to remove any potential effects due to age/maturity (i.e. Parity) and condition of the cow (i.e. iBW). This analysis of identifying the "best" and

"worst" performing animals was done for each of the three windows periods, which resulted in different sets of selected animals, depending on the window period. In order to identify which of the three periods better expressed the impact of FT on performance, two additional analyses were performed. First, the residual variance ($\sigma_e^2$) of the data for each of the window periods were estimated with the model below, in order to identify the period in which greater variability of the data was observed, which is an indication of response to diseases [22]:

$$AWG_{ijk} = \mu + L_i + P_j + b_1(iBW_k - \overline{iBW}) + e_{ijk} \qquad \text{[Eq 2]}$$

where $AWG_{ijk}$, $\mu$, $L_i$, $P_j$, $b_1$, $iBW_k$, and $e_{ijk}$ are as previously defined in Eq 1, assuming $e_{ijk} \sim N(0, \sigma_e^2)$. Analysis was performed in SAS 9.4. The estimated $\sigma_e^2$ of each window period (w1_7, w1_13, and w7_13) was compared between each other and tested using an F-test. In addition, the AWG residuals (AWG_res) of the selected animals based on Eq 1 were analyzed with the following model:

$$AWG\_res_{ijk} = \mu + T_i + L_j + W_k + interactions + e_{ijk} \qquad \text{[Eq 3]}$$

where $\mu$ and $e_{ijk}$ are as previously defined, assuming $e_{ijk} \sim N(0, \sigma_e^2)$; $AWG\_res_{ijk}$ is the AWG_res of the selected animal; $T_i$ is the fixed effect of the $i^{th}$ tolerance group (HT or LT), $L$ is the fixed effect of the $j^{th}$ location (BBFCL or UPRS); $W_k$ is fixed effect of the $k^{th}$ window period (w1_7, w1_13, or w7_13); and *interactions* represent all possible interactions between these effects.

## Quantification of alkaloid concentrations

Fescue tiller samples were collected in November of 2016 to evaluate pasture infection rate for the toxic endophyte. Fescue tiller samples were collected on a particular day, rinsed the following evening, and shipped on ice the following morning to determine pasture infection rate and the average infection rate is reported by experimental period (Agrinostics Ltd. Co., Watkinsville, GA). Fescue samples from each pasture were sent to the University of Missouri Veterinary Medical Diagnostic Laboratory (Columbia, MO) to analyze the ergot alkaloid amounts present within the grass using HPLC as described by Rottinghaus et al., [23].

## DNA extraction

Fecal material from the selected 40 HT and LT cows was thawed and genomic DNA was extracted from 0.25 grams of sample, using the Qiagen DNeasy Powerlyzer Powersoil kit following the instructions of the manufacturer. Mechanical cell lysis was performed using a Fischer Scientific Beadmill 24. DNA concentrations were determined using a Qubit 3 fluorometer (Invitrogen).

## Sequencing and analysis

**16S rRNA gene.** Briefly, PCR amplicon libraries targeting the 16S rRNA gene present in extracted DNA were produced using a barcoded primer set adapted for Illumina MiSeq [24]. DNA sequence data was generated using Illumina MiSeq paired-end sequencing at the Environmental Sample Preparation and Sequencing Facility (ESPSF) at Argonne National Laboratory (Lemont, IL, USA). Specifically, the V4 region of the 16S rRNA gene (515F-806R) was PCR amplified with region-specific primers that include sequencer adaptor sequences used in the Illumina MiSeq flowcell [24, 25]. The forward amplification primer also contains a twelve base barcode sequence that supports pooling of up to 2,167 different samples in each lane [24, 25]. Each 25 μL PCR reaction contained 9.5 μL of MO BIO PCR Water (Certified DNA-Free),

12.5 μL of QuantaBio's AccuStart II PCR ToughMix (2x concentration, 1x final), 1 μL Golay barcode tagged forward primer (5 μM concentration, 200 pM final), 1 μL reverse primer (5 μM concentration, 200 pM final), and 1 μL of template DNA. The conditions for PCR were as follows: 94˚C for 3 minutes to denature the DNA, with 35 cycles at 94˚C for 45 s, 50˚C for 60 s, and 72˚C for 90 s; with a final extension of 10 min at 72˚C to ensure complete amplification. Amplicons were then quantified using PicoGreen (Invitrogen) and a plate reader (Infinite® 200 PRO, Tecan). Once quantified, volumes of each of the products were pooled into a single tube so that each amplicon was represented in equimolar amounts. This pool was then cleaned up using AMPure XP Beads (Beckman Coulter), and then quantified using a fluorometer (Qubit, Invitrogen). After quantification, the molarity of the pool was determined and diluted down to 2 nM, denatured, and then diluted to a final concentration of 6.75 pM with a 10% PhiX spike for sequencing on the Illumina MiSeq. Amplicons were sequenced on a 151bp MiSeq run using customized sequencing primers and procedures [24].

Sequence analysis was done with the open source software mothur following the mothur MiSeq SOP [26] using mothur version 1.39.3. Barcode sequences, primer and low-quality sequences were trimmed using a minimum average quality score of 35, with a sliding window size of 50 bp. Chimeric sequences were removed with the "Chimera.uchime" command. For alignment, the SILVA SSU NR reference database v128 [27] and for taxonomic classification the SILVA SSU NR v138 database were used. After quality control, 58,400 sequences were randomly subsampled from each sample using mothur. The sequences were clustered into operational taxonomic units (OTU) with a cutoff of 97% 16S rRNA gene similarity (= 0.03 distance).

**ITS1 region.** Library preparation and amplicon sequencing was performed using Illumina MiSeq sequencing platform as with the 16S rRNA analysis. The ITS1 region was amplified using ITS1f-ITS2 primers designed to amplify fungal microbial eukaryotic lineages designed by the Earth Microbiome Project [28]. This generated paired-end reads of 251bp. Sequence analysis was done with mothur as for 16S rRNA genes. Barcode sequences, primers and low quality sequences were trimmed using a minimum average quality score of 35, with a sliding window size of 50bp. Sequences were aligned against themselves using the mothur command "pairwise.seqs", and the UNITEV8_sh_99 dataset provided by UNITE [29] was used to classify the sequences. After quality control, 10,000 sequences were randomly subsampled from each sample using mothur. The sequences were clustered into OTUs with a cutoff of 97% ITS1 region similarity (= 0.03 distance) following recent guidelines [30]. Additionally, representative sequences for each OTU were further classified using NCBI BlastN.

## Statistical analysis

The amplicon sequencing data was analyzed with the same model described in Eq 2. Prior to analyses, the OTU data was normalized using trimmed mean of M values (TMM; [31]), which has been shown to work well using microbiome data [32]. Normalization was carried out in R [33] using the *edgeR* package [34]. In this analysis, the 50 most abundant OTUs for each dataset (16S rRNA and ITS1) were analyzed. The data were then adjusted for multiple comparison using false-discovery rate (FDR; [35]) using the *qvalue* [36] package in R.

Preliminary analyses indicated statistical problems with the ITS1 data because of the low counts for some OTUs. Therefore, OTUs with low counts (*n* = 4) in each location-tolerance group combination were removed. Means (for the diversity data) and log2 fold-changes (log2FC; for the OTU count data) were separated using Tukey's test for the effects of location, tolerance group, and their interaction, when significant (*q<0.05*). Tables containing the log2FC and confidence intervals for all effects per OTU as well as the test results and the FDR corrections (q-values) are presented in S1 and S2 Tables. In addition to these analyses, a

canonical discriminant analysis (CDA) was performed with the objective of identifying OTUs (both 16S rRNA and ITS1 data) that could discriminate groups with high power. Three CDA analyses were performed to discriminate between tolerance groups (2 groups), locations (2 groups), and between the 4 groups from the combination between location and tolerance group. OTUs were selected using a stepwise approach, with an alpha of 0.15 to enter the model and an alpha of 0.05 to remain in it. In addition, a leave-one-out cross-validation (LOOCV) was performed in order to assess the classification power of OTU. This analysis was done using the relative abundance data because of its more quantitative characteristics. All statistical analyses were performed in SAS 9.4 (SAS Institute Inc., Cary, NC).

## Results

### Animal selection based on modeling of AWG

When testing the main effects and interactions, no significant effects ($P \geq 0.159$) for the main effects of $W$ and $L$, and for the interactions of $T^*L^*W$, $L^*W$, and $T^*L$ were found. There was a significant ($P<0.0001$) interaction between $T$ and $W$.

The estimated $\sigma_e^2$ for each window period and for each T by window period are presented in S3 Table. The estimated $\sigma_e^2$ for w1_7 [6.00 (kg/week)$^2$] was greater ($P<0.01$) than for w1_13 [0.07 (kg/week)$^2$] and w7_13 [2.94 (kg/week)$^2$]. In addition, HT animals for w1_7 had the highest ($P<0.01$) AWG_res (3.74 kg/week), whereas LT animals for w1_7 (-3.52 kg/week) had lowest ($P<0.01$) AWG_res, and all of the other AWG_res were not different from each other ($P>0.01$). Because of the greater estimated $\sigma_e^2$ and the more extreme AWG_res values, data using w1_7 were used for subsequent analyses.

### Ergot alkaloid concentrations per farm determined by HPLC

Overall, the percentage of fescue in the pastures was not significantly different between locations (68.1 and 64.3% at UPRS and BBCFL, respectively) throughout the grazing period. Ergovaline levels were 1,110 µg/Kg and 1,900 µg/Kg were found at BBCFL and UPRS farms respectively (Table 1). The UPRS farm showed higher ergot alkaloid concentrations than the BBCFL farm, harboring in addition to Ergovaline also Ergosine, Ergotamine, Ergocornine, Ergocryptine and Ergocristine.

### Composition of fecal bacterial microbial communities

Overall, 5,320 OTUs were generated after quality control, subsampling and removal of OTUs representing less than ten sequences from the original 4.675 million sequence reads, of which

**Table 1. Ergot alkaloid concentration of tall fescue pastures.**

| Ergot alkaloids | Concentration at farm BBCFL[1] (µg/Kg) | Concentration at farm UPRS[2] (µg/Kg) |
|---|---|---|
| Ergosine | 0 | 1,000 |
| Ergotamine | 0 | 525 |
| Ergocornine | 0 | 160 |
| Ergocryptine | 0 | 450 |
| Ergocristine | 0 | 145 |
| Ergovaline | 1,110 | 1,900 |
| Total | 1,110 | 4,180 |

[1]BBCFL: Butner Beef Cattle Field Laboratory, Bahama NC

[2]UPRS: Upper Piedmont Research Station, Reidsville NC

3.983 million reads (85%) remained after quality control. Most of the reads were bacterial, 0.8% of all reads were classified as *Archaea*. Twenty-three phyla were identified with *Firmicutes* (58.7–67.9%), *Bacteroidetes* (19.7–25.7%), *Proteobacteria* (1.1–12.8%), *Actinobacteria* (1.1–3.7%) and unclassified bacteria (5.2–7.7%) being most abundant (S1 Fig). All other phyla showed relative abundances of less than 1%.

The most abundant OTU (OTU 1, comprising 7.6% of all reads) affiliated to the *Oscillospiraceae* UCG-005 group, OTU 2 to *Solibacillus* (99.6% similarity to *Solibacillus silvestris*, 6.1% overall relative abundance), OTU 3 to *Acinetobacter* (100% similarity to *Acinetobacter lwoffii*, 4.7% overall relative abundance), OTU 4 to *Bacillus* (99.2% similarity to *Bacillus psychrosaccharolyticus*, 3.2% overall abundance) and OTU 5 to *Oscillospiraceae* UCG-005 (93.7% similarity to *Monoglobus pectinilyticus*, 2.3% overall abundance) (S4 Table). Among the most abundant OTUs, a number of OTUs were classified into the same genus such as OTUs 1, 12, 16 29 (*Oscillibacter*), 5, 47 (*Monoglobus*); 8, 13, 28, 35 (*Bacteroides*); 9, 14 (*Lysinibacillus*); 2, 46 (*Solibacillus*).

## Composition of fecal fungal microbial communities

Overall, 1,000 OTUs were generated after quality control, subsampling and removal of OTUs representing less than 10 sequences from the original 7.051 million sequence reads, of which 390,000 reads remained after quality control and subsampling.

OTU 1, OTU 5 and OTU 12 affiliated to *Microsphaeropsis* (Montagnulaceae family), OTU 2, 3, 7, 8, 10, 13, 16, 17, 18, 40, 46 to *Thelebolus* (Thelebolaceae family), OTU 4 to the Pleosporaceae family, OTUs 14, 22, 34, 35 to *Orpinomyces* and OTUs 25, 26, 31, 45 to *Caecomyces*; both of these genera belong in the family Neocallimastigaceae (S5 Table). No sequences were attributed to *Epichloë coenophiala*, the endophyte believed to be responsible for FT, by either the classification against the UNITE database or manual BlastN of representative sequences of each OTU against NCBI nr.

## Alpha diversity of HT and LT cattle fecal microbial communities

We observed statistically significant differences when comparing alpha diversity metrics of the bacterial microbial communities for the effect of tolerance and the T*L interaction. We found that species richness estimators Chao, and ACE reported a significant (*P<0.019* and *P<0.025*, respectively) interaction effect (Table 2). Although this may be due to differing alkaloid composition and concentrations between farms (see Table 1), this may also be affected by different

**Table 2. Bacterial species richness and diversity estimators in fecal microbial communities across location[1] and tolerance groups[2].**

| Diversity Parameter | BBCFL | | UPRS | | *P*-value[3] | | |
|---|---|---|---|---|---|---|---|
| | HT | LT | HT | LT | T | L | T*L |
| Ace (richness) | 9473.9[b] (557.9) | 9245.1[b] (557.9) | 12074[a] (557.9) | 9187.8[b] (588.0) | 0.009 | 0.031 | 0.025 |
| Chao (richness) | 6620.9[b] (318.7) | 6506.2[b] (318.7) | 8127.3[a] (318.7) | 6418.1[b] (335.9) | 0.008 | 0.035 | 0.019 |
| Npshannon (diversity) | 5.80[a] (0.16) | 5.29[b] (0.16) | 5.98[a] (0.16) | 5.06[b] (0.17) | <0.001 | 0.918 | 0.209 |
| Shannon (diversity) | 5.71[a] (0.17) | 5.20[b] (0.17) | 5.90[a] (0.17) | 4.97[b] (0.17) | <0.001 | <0.889 | 0.213 |
| Simpson (evenness) | 0.018[bc] (0.014) | 0.057[ab] (0.014) | 0.017[c] (0.015) | 0.062[a] (0.014) | 0.005 | 0.9 | 0.85 |

[1]BBFLC, Butner Beef Cattle Field Laboratory (Bahama, NC, USA); UPRS, Upper Piedmont Research Station (UPRS; Piedmont, NC, USA)

[2]HT, High Tolerance; LT, Low Tolerance

[3]P, Tolerance group (HT or LT); L, Location (BBCFL or UPRS); L*T, interaction between T and L

[a,b,c] Least-squares means of alpha diversity values lacking common superscripts are statistically different (*P<0.05*) based on Tukey's test

Numbers within parentheses represent standard error measurements.

**Table 3. Fungal species richness and diversity estimators for fecal microbial communities across location[1] and tolerance groups[2].**

| | BBCFL | | UPRS | | *P*-value[3] | | |
|---|---|---|---|---|---|---|---|
| Diversity Parameter | HT | LT | HT | LT | T | L | T*L |
| Ace (richness) | 4970.96[b] (1405.16) | 7282.02[ab] (1405.16) | 10593[a] (1405.16) | 9697.22[a] (1481.17) | 0.623 | 0.008 | 0.268 |
| Chao (richness) | 2732.72[b] (544.6) | 3270.42[b] (544.6) | 5256.33[a] (544.6) | 4026.99[ab] (574.06) | 0.535 | 0.005 | 0.119 |
| Npshannon (diversity) | 3.27[ab] (0.28) | 2.86[b] (0.28) | 4.05[a] (0.28) | 2.48[b] (0.3) | 0.001 | 0.484 | 0.05 |
| Shannon (diversity) | 3.1[ab] (0.27) | 2.69[bc] (0.27) | 3.82[a] (0.27) | 2.23[c] (0.27) | 0.001 | 0.653 | 0.039 |
| Simpson (evenness) | 0.151[b] (0.06) | 0.272[ab] (0.06) | 0.132[b] (0.06) | 0.372[a] (0.062) | 0.005 | 0.501 | 0.325 |

[1]BBCFL, Butner Beef Cattle Field Laboratory (Bahama, NC, USA); UPRS, Upper Piedmont Research Station (UPRS; Piedmont, NC, USA)

[2]HT, High Tolerance; LT, Low Tolerance

[3]T, Tolerance group (HT or LT); L, Location (BBCFL or UPRS); L*T, interaction between T and L

[a,b,c] Significant differences in alpha diversity values between diversity parameter and effect (T, L, T*L) are designated by lowercase letters (*P<0.05*)

Numbers within parentheses represent standard error measurements

management practices or local climate differences between the locations. In addition, we observed significant decreases in diversity (Shannon, *P<0.001*) and species richness (Chao, *P = 0.0078*; ACE, *P = 0.0093*) and an increase in evenness (Simpson, *P = 0.005*) in the LT cattle for both sites strictly due to tolerance group.

Comparing alpha diversity metrics of the fungal microbial communities for the effect of tolerance, location, and the T*L interaction revealed significant differences as well (Table 3). Shannon and NpShannon were significantly affected by a T*L interaction effect, but also by tolerance. Similar to above, this may be due to general differences between the two farms. Species richness estimators Chao, and ACE were also significantly (*P<0.008* and *P<0.005*, respectively) affected by location. Finally, as with the 16S rRNA data, a significant increase in species evenness (Simpson, *P = 0.0047*) was observed in LT cattle from both sites considering the tolerance group effect.

## Differentially abundant OTUs between HT and LT cattle

Among the 50 most abundant OTUs, we observed statistically significant (*q<0.05*) differences in abundance between tolerance and location groups, as well as significant interactions between them. When considering interactions between location and tolerance group for the 16S rRNA, OTUs 6, 15, 18, 19, 20, 26, 27, 38, and 45 showed significant (*q<0.05*) interactions, although with opposite effects for each location. For the fungal OTUs, eleven OTUs were significant (*q<0.05*) for the interaction effect. Out of which, three OTUs shared similar effects at each location, although to a differing degree (thus the significant interaction effect). By sharing the same effect due to location, we can interpret and discuss the "nested" effect tolerance within the interaction effect for these three OTUs: OTUs 5 (*Microsphaeropsis*) and 6 (*Psilocybe*), which were more abundant in the HT cattle, and OTU17 (*Thelebolus*) which was more abundant in the LT cattle at both farms (Fig 1).

Seven bacterial and nine fungal OTUs were significantly different when considering only the main effect of location for comparison (Fig 2). Though recorded here, they might not have relevance to the biological question studied in this manuscript.

Considering the main effect of tolerance group, bacterial OTU 3 (*Acinetobacter*) was significantly (*q<0.0001*) higher in HT cattle and eleven fungal OTUs were significantly (*q<0.05*) different between HT and LT cattle (Fig 3). Fungal OTUs 2, 3, 13 (all three classified as *Thelebolus*) were more abundant in LT cattle, whereas fungal OTUs 1 (*Microsphaeropsis*), 14

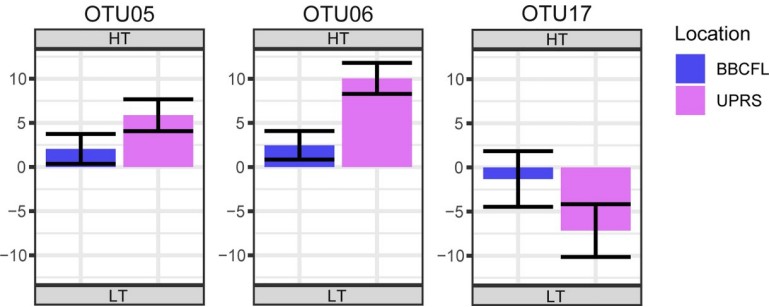

**Fig 1. Statistically significantly ($q \leq 0.05$) different abundant fungal OTUs considering interactions of location (BBCFL, UPRS) and tolerance group (HT, LT) as an effect where effects due location had the same direction.** The differences in abundance of fungal OTUs are shown as log2 fold changes for each tolerance interaction. Positive values represent higher abundance in HT cattle, negative values represent higher abundance in LT cattle. Bars in blue and purple represent farms BBCFL and UPRS, respectively. Error bars represent the 95% confidence interval.

and 22 (*Orpinomyces*), 20 (*Pyrenochaetopsis*), 26 and 31 (*Caecomyces*), and 38 and 49 (*Pleosporales*) were more abundant in HT cattle.

## Community-level comparison of microbial communities using canonical discriminant analysis

Results of the CDA analyses are presented in Table 4 and depicted in Fig 4 and S2 Fig. There were 8, 19, and 14 OTUs selected for the analyses of T, L, and T*L, respectively, with four OTUs overlapping between these (bacterial OTUs 19 (*Rikenellaceae* RC9 gut group) and 21 (uncultured *Ruminococcaceae*) and fungal OTUs 1 (*Microsphaeropsis*) and 6 (*Psilocybe*). The discrimination of groups based on CDA was significant ($P<0.001$) for all canonical variables (CAN) for both analyses. The squared canonical correlations ($R^2$) were: 94.7% for T, 99.26% for L, and 98.4% (CAN1), 92.5% (CAN2), and 73.0% (CAN3) for L*T. For L*T, the proportion of the total variation explained by each CAN was 80.2% (CAN1), 16.2% (CAN2), and 3.6% (CAN3). For T, the two OTUs showing the most discriminative power were fungal OTU1 (*Microsphaeropsis*) and bacterial OTU21 (unclassified *Ruminococcaceae*), with standardized canonical coefficients (SCC) of -4.0 and 3.1, respectively. For L, these were (SCC in parentheses): bacterial OTU1 (*Ruminococcaceae* UCG-005, -5.3) and OTU2 (*Solibacillus*, 5.2). For T*L, these were: fungal OTU6 (*Psilocybe*, 5.4) and fungal OTU8 (*Thelebolus*, 2.2) for CAN1, fungal OTU25 (*Caecomyces*, -3.0) and OTU1 (*Microsphaeropsis*, 2.9) for CAN2, and fungal OTU25 (*Caecomyces*, 1.5) and bacterial OTU21 (unclassified *Ruminococcaceae*, 1.4) for CAN3. The misclassification rates for the CDA of T, L, and T*L were 2.6%, 0%, and 5.3%, respectively.

## Discussion

In general, our knowledge about FT has significantly advanced during recent years [1, 2, 4, 17, 37–41]. However, the knowledge about a possible involvement of GI tract microbial communities, especially fungal communities, in FT is still highly limited. Recently, Mote et al found in animals fed toxic fescue, relative abundances of the bacterial families *Ruminoccocaceae* and *Lachnospiraceae* in fecal samples were significantly increased [17]. Our study predominantly found consistent changes in the fungal OTUs and only detected a single significantly different bacterial OTU within the 50 most abundant OTUs between the tolerance groups, and this OTU was not related to either of the aforementioned families (Fig 3). It should be noted that the study by Mote et al., used Angus steers, not cows, and was performed at a different location (in Georgia, USA) [17]. Thus, the comparability of these two studies might be limited.

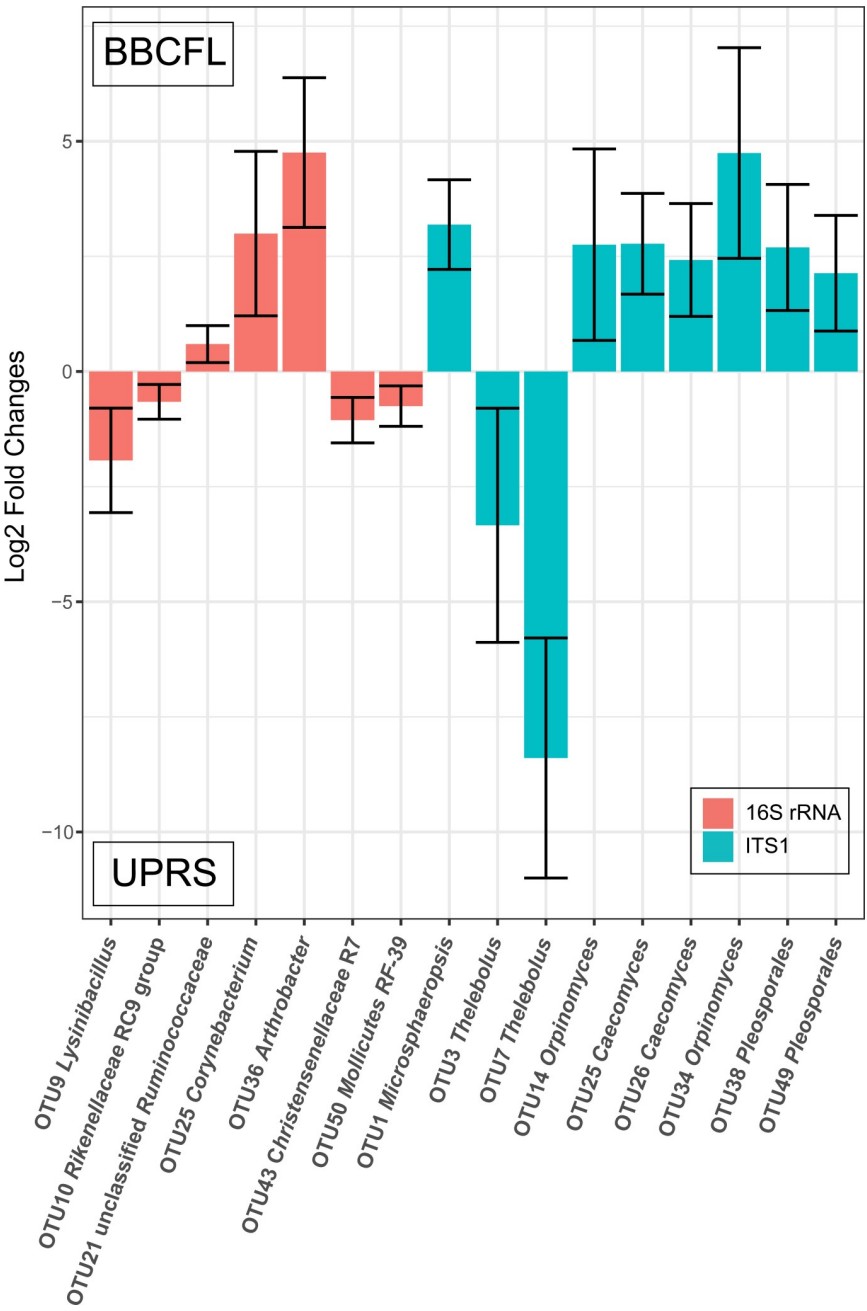

**Fig 2. Statistically significantly ($q \leq 0.05$) different abundant bacterial and fungal OTUs considering location (BBCFL, UPRS) as an effect.** The differences in abundance of OTUs are shown as log2 fold changes for each location. Positive values represent higher abundance in farm BBCFL, negative values represent higher abundance in farm UPRS. Error bars represent the 95% confidence interval.

Particularly, keeping in mind the strong differences in microbiota between farms found here. Nevertheless, similar to our study, Mote et al. also identified major changes of fecal the microbial communities in response to FT [17]. Another study has shown degradation of fescue alkaloids by rumen microorganisms without identifying the microbes responsible for the degradation [19]. Tryptophan-utilizing rumen bacteria can be capable of ergovaline degradation as shown for a *Clostridium sporogenes*, other *Clostridium* species [42], and a *Prevotella*

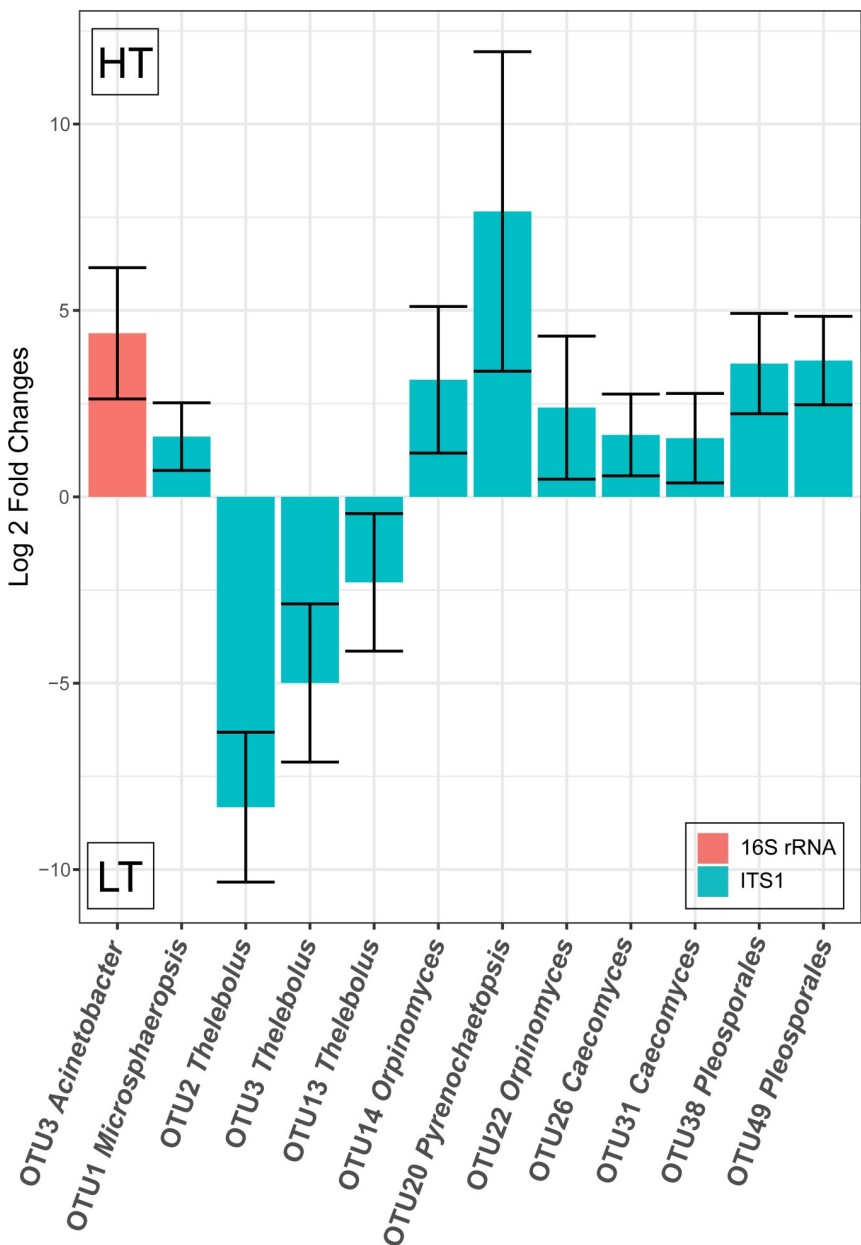

**Fig 3. Statistically significantly ($q{\leq}0.05$) different abundant bacterial and fungal OTUs considering tolerance group (HT, LT) as an effect.** The differences in abundance of OTUs are shown as log2 fold changes for each tolerance group. Positive values represent higher abundance in HT cattle, negative values represent higher abundance in LT cattle. Error bars represent the 95% confidence interval.

*bryantii* isolate [18]. We did not find *Clostridium sporogenes* or *Prevotella bryantii* OTUs in our dataset; Although *Clostridium* OTU 15, was significantly different considering T*L, opposing effects between locations make any judgements of tolerance effect for this OTU difficult without additional clarifying data. Another explanation for the absence of *Prevotella bryantii* and *Clostridium sporogenes*, which are abundant rumen bacteria, in our samples might be that we have used fecal (and not rumen) samples and rumen bacteria might not be well represented in fecal samples. Additionally, a *Rhodococcus erythropolis* strain has recently been described to

**Table 4. Canonical discriminant analysis for tolerance group[1] (T), Location[2] (L), and interaction between T and L (T*L).**

| | | | Standardized Canonical Coefficients | | | | |
| | | | T | L | P*L | | |
| Target | OTU | Classification | CAN1 | CAN1 | CAN1 | CAN2 | CAN3 |
|---|---|---|---|---|---|---|---|
| 16S rRNA gene | OTU01 | *Oscillospiraceae* UCG-005 | | 5.3 | | | |
| | OTU02 | *Solibacillus silvestris* | | 5.2 | | | |
| | OTU05 | *Monoglobus pectinilyticus* | | 2.5 | | | |
| | OTU10 | *Rikenellaceae* RC9 gut group | | 1.3 | | | |
| | OTU12 | *Oscillospiraceae* UCG-005 | | | 1.1 | -0.1 | -0.4 |
| | OTU13 | *Bacteroides plebeius* | | 1.6 | | | |
| | OTU15 | *Clostridium difficile* | 0.8 | | | | |
| | OTU19 | *Rikenellaceae* RC9 gut group | 1.8 | 2.7 | -0.2 | -1.2 | -0.9 |
| | OTU21 | unclassified *Ruminococcaceae* | 3.1 | -1.6 | -0.6 | -2.0 | 1.4 |
| | OTU22 | *Psychrobacillus psychrodurans* | 0.7 | | | | |
| | OTU25 | *Corynebacterium kutscheri* | -2.3 | | | | |
| | OTU30 | *Christensenellaceae* R7 group | | | -0.6 | 1.8 | -0.2 |
| | OTU31 | unclassified *Bacteroidetes* | | -2.3 | -1.2 | 1.5 | 0.1 |
| | OTU34 | *Rikenellaceae* dgA-11 gut group | | -1.3 | | | |
| | OTU39 | *Prevotellaceae* UCG-003 | -1.9 | | | | |
| | OTU40 | *Anaerotignum faecicola* | -1.9 | -4.3 | | | |
| | OTU48 | *Rikenellaceae* RC9 gut group | | 1.5 | 0.7 | 0.5 | -0.3 |
| | OTU50 | *Mollicutes* RF39 | | 1.8 | 0.9 | -0.2 | < -0.1 |
| ITS1 region | OTU01 | *Microsphaeropsis arundinis* | -4.0 | -4.9 | -1.8 | 2.9 | -0.3 |
| | OTU02 | *Thelebolus* | | -1.2 | | | |
| | OTU06 | *Psilocybe* | -2.3 | 1.7 | 5.4 | 1.1 | 0.8 |
| | OTU08 | *Thelebolus* | -1.8 | | 2.2 | 1.1 | <0.1 |
| | OTU12 | *Microsphaeropsis* | | | -0.5 | 2.0 | -0.9 |
| | OTU13 | *Thelebolus* | 0.9 | | | | |
| | OTU19 | *Phaeosphaeriopsis* | | 1.7 | 1.2 | -0.3 | -0.1 |
| | OTU24 | *Coprinopsis cothurnata* | | -3.1 | | | |
| | OTU25 | *Caecomyces* | 1.7 | | <0.1 | -3.0 | 1.5 |
| | OTU31 | *Caecomyces* | | -1.5 | | | |
| | OTU34 | *Orpinomyces* | | | < -0.1 | 1.0 | -0.4 |
| | *P*-value | | <0.001 | <0.001 | <0.001 | <0.001 | <0.001 |
| | $R^2$ (%) | | 94.7 | 99.3 | 98.4 | 92.5 | 73.0 |

[1]BBCFL, Butner Beef Cattle Field Laboratory (Bahama, NC, USA); UPRS, Upper Piedmont Research Station (UPRS; Piedmont, NC, USA)

[2]HT, High Tolerance; LT, Low Tolerance

CAN1-CAN3, canonical variables 1–3. The number of CAN in each analysis depends on the number of levels of the discriminated group (i.e. *n*-1)

degrade ergot alkaloids [21]. We found only very few and extremely low abundant *Rhodococcus* OTUs in our dataset; based on the reported strain-specific ergot alkaloid degradation capability of *Rhodococcus* [21], we assume that *Rhodococcus* species are not involved in ergot alkaloid degradation in the animals analyzed here.

We chose to study the fecal microbiota as fecal samples allow for periodic observations of microbiota with large sample sizes in a non-invasive way. One major limitation of using fecal samples to study GI tract microbiota is the fact that the fecal microbiota primarily reflects luminal microbiota, which can be significantly different from GI tract mucosal microbiota. Moreover, fecal samples are more representative of the digesta in the lower GI tract and do not

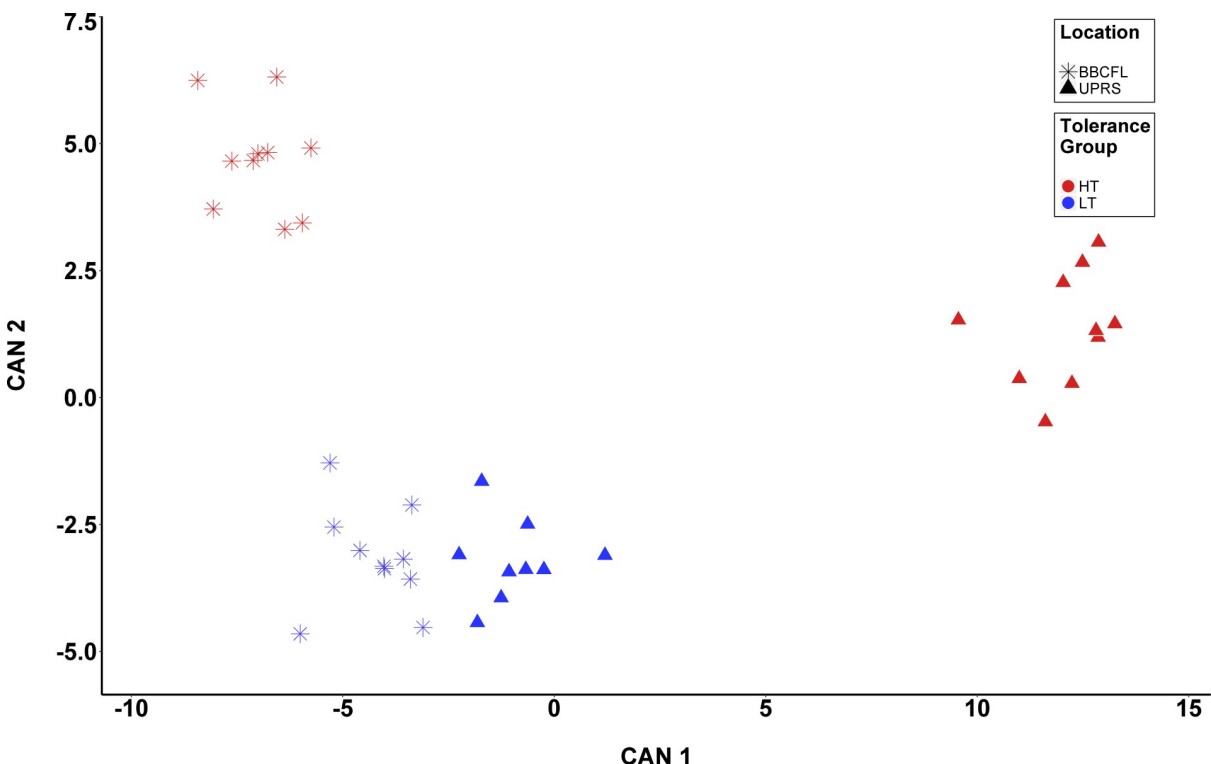

**Fig 4. Canonical discriminant analysis (CDA) for response to fescue toxicosis.** CDA was performed to discriminate animals based on the combination between tolerance group (T; High [HT] and Low [LT] tolerance groups) and location (BBCFL and UPRS) using 14 fungal and bacterial OTUs. Each point represents the canonical score (CS) of each animal based on the respective canonical variable (CAN). The x-axis represents CAN 1 and the y-axis CAN 2. Red and Blue points represent HT and LT animals, respectively, whereas triangles and stars represent animals from UPRS and BBCFL, respectively.

necessarily adequately represent rumen microbial communities. It should thus be noted that the findings found here using fecal samples might not reflect changes of the microbiota of the rumen. In the future, an investigation of microbiota changes in response to toxic fescue feeding should be performed using rumen samples as FT is considered a rumen metabolic disease.

16S rRNA gene amplicon sequencing revealed highly different microbial communities in the HT and LT cattle at both farms analyzed. The shifts in microbial community diversity (Shannon, npShannon) were more pronounced at farm UPRS, where the animals were fed a diet containing higher ergovaline levels and more diverse ergot alkaloids, suggesting a tolerance-by-location interaction (T*L), which is consistent with [43] (Table 1). The pasture at farm UPRS contained approximately 3.6-fold higher overall ergot alkaloids compared to farm BBCFL and, in addition, the UPRS pasture also contained more ergot alkaloids other than ergovaline than BBCFL such as ergosine, ergotamine, and ergocryptine. This may indicate that other ergot alkaloids than ergovaline may be important for the stronger FT observed at farm UPRS. This may also suggest that the amount of alkaloid toxins in the diet determines not only the severity of FT, but also changes in microbial community composition. These hypotheses would need to be verified experimentally in the future. Similarly, on a community level, LT and HT cattle fecal microbial communities showed a significantly different composition as highlighted by the CDA analysis which revealed a clear clustering of fecal communities with respect to location and tolerance group. Also, on OTU-level, significant differences between fecal microbiota were revealed in our study for location as well as for tolerance group.

When considering T*L interactions, for the 16S rRNA data, differing effects on abundance for specific bacterial OTUs were observed for the two farms making it difficult to determine which effect (tolerance or location) has more of an impact on the abundance of these OTUs. When interpreting the significant interaction results for OTU comparisons, we erred on the side of caution and focused on OTUs that shared the same effect direction (both positive or negative log2 fold changes) for location. We believe that these interactions have meanings, and if they shared the same effect direction for each location, it can be interpreted that differences in abundance between tolerance groups are less affected by location. However, the OTUs that did not share a direction of effect for location may have been more heavily influenced by factors related to location, such as management practices, climate or possibly the differing level of alkaloid toxins. These latter OTUs are listed in this study, but with no easy way to account for the interaction effect, they were not considered to have a primary effect on fescue tolerance. These bacteria may derive from the different environments at the two farms, possibly resulting from different feeding and management strategies at both farms. This could suggest that bacteria may not be of key relevance for different response to FT. For the fungal data, three OTUs were found to share the same location effects, albeit to differing degrees, allowing us to draw conclusions of how the tolerance affects the abundance of these OTUs within the context of the interaction. OTUs 5 (*Microsphaeropsis*) and 6 (*Psilocybe*) were more abundant in the HT cattle and OTU 17 (*Thelebolus*) was more abundant in LT cattle, at both farms.

Seven bacterial OTUs and nine fungal OTUs were significantly different when considering only location. Bacterial OTUs 9 (*Lysinibacillus*), 10 (*Rikenellaceae* RC9 gut group), 43 (*Christensenellaceae*), and 50 (*Mollicutes* RF-39*)* were more abundant at farm UPRS and 21 (unclassified *Ruminococcaceae)*, 25 (*Corynebacterium*), and 36 (*Arthrobacter*) were more abundant at farm BBCFL. Fungal OTUs 3 and 7 (*Thelebolus*) were more abundant in UPRS and 1 (*Microsphaeropsis*), 14 and 34 (*Orpinomyces*), 25 and 26 (*Caecomyces*), and 38 and 49 (*Pleosporales*) were more abundant at BBCFL. These differences in abundance may be caused by the different levels of toxins or different management and feeding strategies and feed composition between the two farms.

For tolerance group, only one bacterial OTU (OTU 3, *Acinetobacter lwoffii*) was significantly different and higher in HT cattle. *A. lwoffii* can cause bacteremia in humans [44] and different *Acinetobacter* species have been found in ruminant GI tracts [45]. One study suggested a beneficial role of *A. lwoffii* isolated from cattle shedding on the reduction of allergies [46], while another study has shown a low abundance of *A. lwoffii* in ruminant abortions [47]. It is currently unclear whether these *Acinetobacter* phylotypes found in cattle may be opportunistic human pathogens or are part of the physiological GI tract microbiota. In contrast, eleven fungal OTUs were significantly different between HT and LT cattle when considering tolerance group. Fungal OTUs 1 (*Microsphaeropsis*), 14 and 22 (*Orpinomyces*), 20 (*Pyrenochaetopsis*), 26 and 31 (both classified as *Caecomyces*), and 38 and 49 (*Pleosporales*) were found to be significantly more abundant in HT cattle, whereas OTUs 2, 3, 13 (all three classified as *Thelebolus*)were significantly more abundant in LT cattle.

Some of the abundant fungal OTUs could not be classified to genus level and it is thus hard to speculate about a possible function of these fungal phylotypes. Nevertheless, we assume that OTUs which are more abundant in the HT cattle, could potentially contribute to the better performance indicated by their weight gain observed in the HT cattle. Conversely, the OTUs found in LT cattle may be associated with more severe negative effects of FT. It should, however, be noted that a higher abundance of certain OTUs may not necessarily represent higher or lower metabolic activity and the higher abundance of OTUs might also be due to differential survival during the passage through the GI tract and might result in biased abundances observed in fecal samples. The fact that we observed consistent changes in the abundant fungal

OTUs which were higher in HT cattle at both farms, and that their abundance was consistently higher at the UPRS farm (which is characterized by higher ergot alkaloid toxin levels in the diet) suggests that these fungi may be involved in mitigating the effects of FT in those animals. In spite of the differences caused by performing our animal trial at two different locations, the observation of similar and consistent changes in abundance of certain fungal phylotypes in response to FT, provides more support to the hypothesis that these phylotypes could potentially be positively associated with the higher AWG in HT cattle in our study. It should be noted that this potential beneficial role of fungi in FT tolerance needs to be investigated in more detail in future studies.

Anaerobic fungi of the phylum Neocallimastigomycota are effective fiber degrading organisms in the herbivore gut and have been reported to improve feed intake, feed digestibility, feed efficiency, and daily weight gain and milk production [48, 49]. Of the 50 most abundant OTUs, OTUs 14, 22, 25, 26, 31, 34, 35, and 45 were classified as members of the Neocallimastigomycota. Of these, OTUs 14, 22, 26 and 31 were found to be more prevalent in HT cattle. Although we provide no functional data here, this result adds to the accepted consensus these fungi positively contribute to the ruminant system and underlines the importance of gaining functional data in future research to increase efficiency and overall health in livestock species.

The genus *Thelebolus* was attributed to 11 of the 50 most abundant fungal OTUs. Of these OTUs, OTUs 2, 3, 13, and 17 were found to be significantly more abundant in LT cattle, suggesting a negative effect of *Thelebolus* on ruminant health and performance. Members of the genus *Thelebolus* have been found in ruminant samples before [50]. Although there is little published data on this genus, some recent publications have suggested species within this genus can produce a cytotoxic exopolysaccharide designated as Thelebolan [51] and has been recently tested for its apoptotic effect on cancer cells [52]. It is unknown whether this compound contributes to the negative effects in LT cattle during chronic exposure to toxic fescue, and what conditions allow for increased abundance of this genus. It may be possible that the exopolysaccharide has an apoptotic effect on healthy cells in the GI tract, therefore limiting the absorption of nutrients and reducing integumentary strength within the gut.

OTUs attributed to unclassified Pleosporales were identified within our samples consistent with a recent study that found Pleosporales in the ruminant GI tract [50]. Members of the order Pleosporales including *Microsphaeropsis* have been identified as saprobic, endophytic and pathogenic fungi, and they are often are present within animal dung [53]. It is noteworthy that of the 16 OTUs assigned to the order *Pleosporales* within the 50 most abundant fungal OTUs, six OTUs (1, 20, 38, and 49) were found to be significantly more abundant in HT cattle whereas none were found to be more abundant in LT cattle suggesting a potential–although yet to be verified—beneficial role of these Pleosporales phylotypes during a FT challenge. Similar to the anaerobic fungi of the phylum Neocallimastigomycota, it is important to identify possible functional traits in these microorganisms that could be associated with the positive health of these animals with additional research.

Interestingly, we did not find any fungal OTUs related to *Epichloë coenophiala*. This may either be explained by the absence of *Epichloë coenophiala* in the samples sequenced here, possibly caused by degradation of *Epichloë coenophiala* by rumen microorganisms, or that *Epichloë coenophiala* might not be targeted by the primers used for ITS1 amplicon sequencing, or a different, yet unidentified, fungus might be responsible for the alkaloid production resulting in FT in our study. It is also conceivable that the aerobic nature of *Epichloë* as an endophyte prevents its growth under anaerobic conditions of the mammalian GI tract. In addition, fecal samples were collected in the end of the trial, when endophyte infection levels were lower in the forage, which could contribute to this lack of identification of this fungus.

It may be the case that certain fungal species degrade the alkaloid causing FT, thus decreasing the effect of FT directly. Another explanation may be the cellulolytic and fiber-degrading capabilities of fungi. Fungal phylotypes which are more abundant in HT cattle may be producing higher amounts of absorbable nutrients, thereby compensating for the negative impact of FT on GI tract systems. Many fungi are known to produce bioactive antibacterial compounds and could influence microbial community composition and, indirectly, an animals' response to FT. Members of the genus *Microsphaeropsis*, attributed to three of the 50 most abundant fungal OTUs, including OTU1, are known to produce antimicrobial compounds [54], which could potentially alter microbial community composition.

## Conclusion

This study compared the fecal fungal and bacterial communities of Angus cattle that exhibited contrasting tolerance to fescue toxicosis. Both microbial communities were significantly distinct between the HT and LT cattle after chronic exposure to toxic fescue, and may contribute to the animals' physiological response to FT. HT cattle had more even and diverse fecal microbial communities. Cattle with higher tolerance to FT were associated with higher abundances of anaerobic fungi of the phylum Neocallimastigomycota known to break down cellulose, and uncharacterized members of the *Pleosporales* order. Cattle with lower tolerance to FT were found to have higher abundance of phylotypes within the *Thelebolus* genus. This shift in the GI microbiota was more evident at the UPRS location characterized by higher levels of infected fescue, suggesting a tolerance-by-location interaction. In addition, this was the first study to analyze fungal communities associated with contrasting growing performance under FT in cattle. To better understand the contribution of the microbiota, particularly of fungi, to mitigate FT, functional data using rumen samples will be needed in the future. The availability of such data might allow identifying additional ways to mitigate the negative impact of FT on grazing livestock.

## Supporting information

**S1 Fig. Mean relative abundance of the five most abundant bacterial phyla across all sample sites and groups.** The error bars represent SEM.
(PDF)

**S2 Fig. Canonical discriminant analysis (CDA) for response to Fescue Toxicosis (FT).** For A, CDA was performed to discriminate animals based on location group (L): BBCFL and UPRS can be found in blue and gold, respectively. For A, CDA was performed to discriminate animals based on tolerance group (T): High (HT) and Low (LT) tolerance to FT can be found in green and red, respectively. For A, the x-axis represents the CS for CAN 1 and y-axis represent the density of the CS data.
(PDF)

**S1 Table. Probability values and adjusted FDR (q-values) values for the 50 most abundant bacterial and fungal OTUs for Location effect, Tolerance effect and the interaction (L*T) between the two effects.** Cells shaded grey are statistically significantly (q < 0.05) different for the specific effect.
(PDF)

**S2 Table. Log2 fold change values for the 50 most abundant bacterial and fungal OTUs for all combinations of Location, Tolerance and their interactions (T*L).** Fungal OTUs highlighted in grey demonstrate significant interaction between Tolerance and Location but

share directionality of the Location effect as noted in the manuscript and in Fig 3.
(PDF)

**S3 Table. Residual variance ($\sigma\_e^2$)1 for each window period (WP), and AWG_res2 means for each WP by genetic group (GG).**
(PDF)

**S4 Table. The 50 most abundant 16S rRNA gene OTUs.**
(PDF)

**S5 Table. The 50 most abundant ITS1 OTUs.**
(PDF)

## Author Contributions

**Conceptualization:** Daniel H. Poole, Nick V. L. Serão, Stephan Schmitz-Esser.

**Data curation:** Lucas R. Koester, Daniel H. Poole, Nick V. L. Serão.

**Formal analysis:** Lucas R. Koester, Daniel H. Poole, Nick V. L. Serão, Stephan Schmitz-Esser.

**Funding acquisition:** Nick V. L. Serão.

**Investigation:** Lucas R. Koester, Daniel H. Poole, Nick V. L. Serão, Stephan Schmitz-Esser.

**Methodology:** Lucas R. Koester, Daniel H. Poole, Nick V. L. Serão, Stephan Schmitz-Esser.

**Project administration:** Nick V. L. Serão, Stephan Schmitz-Esser.

**Resources:** Daniel H. Poole, Nick V. L. Serão.

**Software:** Lucas R. Koester, Nick V. L. Serão.

**Supervision:** Stephan Schmitz-Esser.

**Visualization:** Lucas R. Koester, Nick V. L. Serão, Stephan Schmitz-Esser.

**Writing – original draft:** Lucas R. Koester, Daniel H. Poole, Nick V. L. Serão, Stephan Schmitz-Esser.

**Writing – review & editing:** Lucas R. Koester, Daniel H. Poole, Nick V. L. Serão, Stephan Schmitz-Esser.

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
