## [Decision Letter · Decision Letter 0]

26 Mar 2020

PONE-D-20-02787

Beef cattle that respond differently to fescue toxicosis have distinct gastrointestinal tract microbiota

PLOS ONE

Dear Dr. Schmitz-Esser

Thank you for submitting your manuscript to PLOS ONE. After careful consideration, we feel that it has merit but does not fully meet PLOS ONE’s publication criteria as it currently stands. Therefore, we invite you to submit a revised version of the manuscript that addresses the points raised during the review process.

We would appreciate receiving your revised manuscript by May 15th. To enhance the reproducibility of your results, we recommend that if applicable you deposit your laboratory protocols in protocols.io, where a protocol can be assigned its own identifier (DOI) such that it can be cited independently in the future. For instructions see: http://journals.plos.org/plosone/s/submission-guidelines#loc-laboratory-protocols

We look forward to receiving your revised manuscript.

Kind regards,

Marcio de Souza Duarte

Academic Editor

PLOS ONE

Reviewers' comments:

Reviewer's Responses to Questions

**Comments to the Author**

1. Is the manuscript technically sound, and do the data support the conclusions?

Reviewer #1: No

Reviewer #2: Partly

2. Has the statistical analysis been performed appropriately and rigorously? 

Reviewer #1: No

Reviewer #2: N/A

3. Have the authors made all data underlying the findings in their manuscript fully available?

Reviewer #1: Yes

Reviewer #2: Yes

4. Is the manuscript presented in an intelligible fashion and written in standard English?

Reviewer #1: Yes

Reviewer #2: Yes

5. Review Comments to the Author

Reviewer #1: General comments:

In this study, the authors evaluated the composition of fecal bacterial and fungal communities of cattle showing distinct growth performance when exposed to toxic fescue. The aim was to investigate microbial groups potentially associated with to fescue toxicosis tolerance. There are several questions regarding the accuracy of the data presented in this manuscript that needs attention. Normalization of the microbial datasets was not described and alignment of the bacterial and fungal sequences has been done using old versions of the Silva and UNITE databases. These are critical steps for the analyses of microbial communities. In addition, FT tolerance does not appear to be a stable phenotype in the animals under study (major changes according to the window period, variations in potentially relevant OTUs according to geographic location), which makes it difficult to assess the true relevance of the fecal microbiota to the FT tolerance phenotype.

Specific comments:

L43: Suggest rephrasing this statement since it is not possible to predict the functional role of the gastrointestinal microbiota simply based on the analysis of fecal samples.

L43: the same animals or the same breed of cattle? Please clarify.

L107-108: It is not clear to the reader how these pastures were evaluated for endophyte infection.

L148: Based on this statement and the results presented in the manuscript, it seems that the FT tolerance is not a stable phenotype, both over time and geographically.

L171-177: These sentences should be moved to the results section.

L215 and L229: The 16S rRNA gene reads were aligned to SILVA SSU NR database v128, which is not the most updated and improved version of the database. The current version 138 has increased considerably the number of available SSU sequences (>9,400,000). The same was done with the fungal ITS1 reads (current version of the UNITE database has >714,000 fungal sequences). Therefore, I recommend updating the taxonomic assignment using the recent version of the Silva and the UNITE databases.

L211-219 and L221-233: Normalization is a critical step during the analysis of microbial datasets that may determine the following statistical analysis as well as the accuracy of the results. The authors should explain here how they normalized their data for bacterial and fungal communities.

L243-245: it is not clear if the problems with the ITS1 data occurred in all samples of if this was observed for only a few OTUs in some animals. Could a normalization step solve these problems? Please clarify.

L262-263 and L299-302: it is not clear how this was evaluated in this study. Importantly, sequences of the endophyte fungus (at genus level) associated with fescue toxicosis have not been identified in the microbial community of fecal samples. Could the authors elaborate on this?

L467: The fact that species of Acinetobacter represent major human pathogens and its potential shedding by animals in the HT group is a matter of concern that should be discussed here.

L507 an L610: The concept of dysbiosis is complex and I suggest removing the term in these sentences, as it may not properly apply to the observed phenotype.

L523-526: Please avoid going back to the same result throughout the discussion, as this may confuse the reader.

L543-545: More abundant OTUs are not necessarily the most active ones, especially in this case, where fecal samples are being investigated and results are being correlated with host performance. If any of these taxa survive better during the passage through the GI tract and more DNA is found in the feces, they will be overrepresented in the microbial community.

Figure 3: to evaluate OTU interactions, it is strongly recommended to apply correction for multiple testing in the microbial datasets and see if these results remain statistically significant.

Table 1: Could the authors comment in their discussion about the differences in the ergot alkaloid concentration of the tall fescue pastures between the two farms under study?

Tables S1 and S2 are of little information to the reader in its current format. Please consider showing the OTU abundance according to each treatment.

Reviewer #2: The objective of this study was to evaluate the fecal microbial communities (bacterial and fungal) from cattle with contrasting growth performance on tall fescue pasture infected by ergot alkaloids. The idea of the study is interesting but I have some major concerns with this study that needs to be addressed. First when looking at the feces you can not make inferences about the ability of the rumen microbiota to metabolize the ergot alkaloids. You need to focus on the feces microbiota as a biological marker associated with cattle with higher tolerance to fescue toxicosis. Second, the way animals that are more tolerant were selected is a strong limitation of this study as many other factors my influence the AWG of cows, which were not considered or controlled. Third, the authors do not understand well hoe to report and discuss interactions. If there is an interaction of tolerance x location you can not report and discuss the effect of tolerance alone as this will be different according to location. This will affect the results and discussion of this paper and need to be corrected.

Ln53: Ergovaline is important and receives more attention but other alkaloids may be just as important in FT. I do not think we have any work that clearly shows that is just ergovaline that is responsible for FT.

Ln81-82: change to: Studies published on the effect of toxic tall fescue on gastrointestinal (GI) microbiota are still limited.

Ln86-86: not “alleviate some of the impact of FT symptoms” but …reduce the toxic effects of the alkaloids and consequent reduce FT…. or something like that.

Ln87-88: what about the rumen protozoa population? Can they have any impact?

Ln89: If the microbes capable of degrading the alkaloids are in the rumen, why are you focussing on the feces and not on the rumen microbial populations?

Ln108-110: suggest changing it to: “Cattle were managed in a rotational grazing system and were moved to a new paddock every two weeks at each location to ensure adequate forage management as well as sufficient forage availability to the cows.”

Line118: …as described by Rottinghaus et al [22].

Ln129-138: I wonder how days of gestation would affect those values. What was the variability in gestation days of those cows? Why wasn’t body condition score considered in the model? AWG may not be the best way to assess performance or resistant to FT of those cows. If they were growing steers or heifers yes.

Ln256: Change to: All statistical analyses were performed using SAS 9.4 (SAS Institute Inc., Cary, NC).

Ln306-307: you can say there was a decreased species richness (Chao, P=0.0078; ACE, P=0.0093) in the LT cattle for both sites. Because there was as treatment x location interaction. When that happens, you have to report the result and discuss the interaction and not the individual treatment effect.

Ln310-312: The same thing here. Need to focus on the interaction when that is significant.

387-388: …between groups of HT and LT cattle were observed for…

Ln396-402: I suggest changing the description of OUT numbers throughout the whole manuscript to their classification. The reader does not know and need to know what is OTU 1 or 2 or 3 and so on. Change this to the classification of each OUT as described in you materials and methods. Again, if there is an interaction you cannot report results of the main treatment alone. This needs to be re-written in the whole manuscript as the way is written it is wrong and confusing.

Ln:495: delete “e.g.”

Ln587-593: can it be that Epichloë coenophiala was just digested by the ruminal microbiota? Why would you expect to find it in feces? Lots of things are happening before it reaches the feces.

6. PLOS authors have the option to publish the peer review history of their article (what does this mean?). If published, this will include your full peer review and any attached files.

Reviewer #1: No

Reviewer #2: No

---

## [Author Response · Author response to Decision Letter 0]

10 Jun 2020

Reviewer #1: 

General comments:

In this study, the authors evaluated the composition of fecal bacterial and fungal communities of cattle showing distinct growth performance when exposed to toxic fescue. The aim was to investigate microbial groups potentially associated with to fescue toxicosis tolerance. There are several questions regarding the accuracy of the data presented in this manuscript that needs attention. Normalization of the microbial datasets was not described and alignment of the bacterial and fungal sequences has been done using old versions of the Silva and UNITE databases. These are critical steps for the analyses of microbial communities. In addition, FT tolerance does not appear to be a stable phenotype in the animals under study (major changes according to the window period, variations in potentially relevant OTUs according to geographic location), which makes it difficult to assess the true relevance of the fecal microbiota to the FT tolerance phenotype.

AU: The data were normalized and we clarified this in the text and below.

With regards to the definition of the FT phenotype, we answered reviewer #2 with a comprehensive explanation. In summary, to the best of our knowledge, no other studies have investigated tolerance to FT. We are proposing a model, and as all models, our approach is not perfect. However, we have provided a potential model to identify FT tolerant animals. We hope our explanation satisfies both reviewers. 

Specific comments:

L43: Suggest rephrasing this statement since it is not possible to predict the functional role of the gastrointestinal microbiota simply based on the analysis of fecal samples.

AU: We agree and deleted this sentence.

L43: the same animals or the same breed of cattle? Please clarify.

AU: We are not sure to what the reviewer refers to here (maybe there is a typo in the line number?): We had mentioned the breed of cattle (Black Angus) in the original manuscript already (L102 in the original manuscript).

L107-108: It is not clear to the reader how these pastures were evaluated for endophyte infection.

AU: We are not sure to what the reviewer refers to here. We had explained the methodology for determining the toxin levels in lines 112-118 in the original manuscript. For increased clarity, we now present the data describing the toxin quantification in a separate paragraph in the methods section.

L148: Based on this statement and the results presented in the manuscript, it seems that the FT tolerance is not a stable phenotype, both over time and geographically.

AU: Thanks for the comment. Although there is still limited information on the immune response of animals to FT, like in many other types of stress (e.g. viral infection, heat stress, etc), the way that animals respond to these stressors is highly time-dependent. Therefore, in our study we wanted to identify what was the time of stress that mostly impacted the response to FT. In fact, we have been successful in using this approach before in studies about host response to diseases (Serão et al., 2016 Genet Sel Evol.; Waide et al., 2018 Genet Sel Evol.; Sanglard et al., 2020 Sci Rep). Differently than using a standard phenotype, such as growth under ideal conditions, the study of stress-related phenotypes are heavily influenced by the time and levels of exposure, as well as the time of the collection of data. Hence, the identification of the time in which animals showed the largest variability in response to FT would indicate which data should be used to measure response to FT. In addition, it is not expected that an attenuated exposure, such as for one of the farms used in this study, would result in the same variability of FT response in these animals. As we added as one of our references (Berghof et al, 2019), a larger variance indicates stress exposure greater than in a “clean” environment. We hope that there reviewer accepts our explanation.

L171-177: These sentences should be moved to the results section.

AU: changed as suggested

L215 and L229: The 16S rRNA gene reads were aligned to SILVA SSU NR database v128, which is not the most updated and improved version of the database. The current version 138 has increased considerably the number of available SSU sequences (>9,400,000). The same was done with the fungal ITS1 reads (current version of the UNITE database has >714,000 fungal sequences). Therefore, I recommend updating the taxonomic assignment using the recent version of the Silva and the UNITE databases.

AU: As suggested, we have now classified the bacterial OTUs against the most recent SILVA database (version 138). We have modified the methods section accordingly. No major differences in classification of the OTUs were observed as a result of using the SILVA 138 version. For those OTUs, where the classification by SILVA v138 yielded different results, the taxonomy was adjusted accordingly. Similarly, the fungal OTUs were classified against the most recent version of the UNITE reference database (V8). Also here, no major changes in the classification of the OTUs were observed.

L211-219 and L221-233: Normalization is a critical step during the analysis of microbial datasets that may determine the following statistical analysis as well as the accuracy of the results. The authors should explain here how they normalized their data for bacterial and fungal communities.

AU: Thanks for pointing this out. The data were normalized using the trimmed mean of M values (TMM) method prior to the final analyses. We added a sentence in the “statistical analysis” section (L229-231 in the revised manuscript).

L243-245: it is not clear if the problems with the ITS1 data occurred in all samples of if this was observed for only a few OTUs in some animals. Could a normalization step solve these problems? Please clarify.

AU: These issues occurred only for a few OTUs in some animals. As mentioned above, data were now normalized using TMM as described in the revised manuscript (Please see also our response above).

L262-263 and L299-302: it is not clear how this was evaluated in this study. Importantly, sequences of the endophyte fungus (at genus level) associated with fescue toxicosis have not been identified in the microbial community of fecal samples. Could the authors elaborate on this?

L262-263: We have deleted this sentence as we deemed the information on percentage of infected fescue not relevant for the revised manuscript.

L299-302: Regarding the lack of detection of Epichloe, we would like to mention that we discussed possible reasons for not being able to detect Epichloe sequences in the discussion section of our manuscript (L587-594 in the original manuscript, L610-618 in the revised version of the manuscript).

L467: The fact that species of Acinetobacter represent major human pathogens and its potential shedding by animals in the HT group is a matter of concern that should be discussed here.

AU: Indeed, OTU3 is classified as Acinetobacter and shows highest similarity to Acinetobacter lwoffii (100% similarity). While the role of A. baumanii as a human pathogen has been well established, the role of A. lwoffii in human disease is less clear. Some evidence suggests that A. lwoffii can cause bacteremia in humans. Different Acinetobacter species have been isolated from cattle and other ruminants. It is currently unknown if cattle Acinetobacter are part of the physiological microbiota of cattle or represent opportunistic pathogens for humans. We had discussed the potential implications of Acinetobacter as part of the bovine fecal microbiota in our original manuscript (L523-526). We have expanded the discussion on Acinetobacter in cattle to accommodate the reviewer’s comment (L550-556 in the revised version of the manuscript).

L507 an L610: The concept of dysbiosis is complex and I suggest removing the term in these sentences, as it may not properly apply to the observed phenotype.

AU: changed as suggested

L523-526: Please avoid going back to the same result throughout the discussion, as this may confuse the reader.

AU: We have deleted the first mentioning of Acinetobacter OTU3 in the revised manuscript and thus refer to Acinetobacter OTU3 only once in the discussion (L550-556 in the revised manuscript).

L543-545: More abundant OTUs are not necessarily the most active ones, especially in this case, where fecal samples are being investigated and results are being correlated with host performance. If any of these taxa survive better during the passage through the GI tract and more DNA is found in the feces, they will be overrepresented in the microbial community.

AU: We agree that abundance does not necessarily reflect metabolic activity. As suggested, we have modified the discussion to mention this limitation (L566-570 in the revised manuscript).

Figure 3: to evaluate OTU interactions, it is strongly recommended to apply correction for multiple testing in the microbial datasets and see if these results remain statistically significant.

AU: Thanks for the comment. We performed a false discovery rate (FDR; Storey, 2002) adjustment. Results are now presented as q-values instead of P-values. A description was added to the text (L232-233 in the revised manuscript).

Table 1: Could the authors comment in their discussion about the differences in the ergot alkaloid concentration of the tall fescue pastures between the two farms under study?

AU: As suggested, we expanded the discussion in the revised manuscript regarding the different alkaloid levels and different ergot alkaloids found in the 2 farms in our study. Please see L516-526 in the revised manuscript.

Tables S1 and S2 are of little information to the reader in its current format. Please consider showing the OTU abundance according to each treatment.

AU: As suggested, we have modified the supplementary Tables showing the 50 most abundant OTUs so that the relative abundance is shown for HT and LT cattle.

Reviewer #2: 

The objective of this study was to evaluate the fecal microbial communities (bacterial and fungal) from cattle with contrasting growth performance on tall fescue pasture infected by ergot alkaloids. The idea of the study is interesting but I have some major concerns with this study that needs to be addressed. First when looking at the feces you can not make inferences about the ability of the rumen microbiota to metabolize the ergot alkaloids. You need to focus on the feces microbiota as a biological marker associated with cattle with higher tolerance to fescue toxicosis. Second, the way animals that are more tolerant were selected is a strong limitation of this study as many other factors my influence the AWG of cows, which were not considered or controlled. Third, the authors do not understand well hoe to report and discuss interactions. If there is an interaction of tolerance x location you can not report and discuss the effect of tolerance alone as this will be different according to location. This will affect the results and discussion of this paper and need to be corrected.

AU: Thanks for your comments. With regards to the classification of cows as high or low tolerant to FT, we added a comprehensive explanation below as well have modified the text.

With regards to the interactions, we partially agree with the reviewer. First, we have changed the way that results are presented to first show the interaction results and then the main effects. For the main effects, we presented results for significant OTUs that did not show an interaction. However, we also presented the results for the OTUs that showed simultaneously significant interactions and main effects only when the direction of the effect was the same. By doing this, we are simultaneously acknowledging that the location/level of exposure plays a role in the OTU abundance between tolerance groups (i.e. interaction) and that, more generally, the difference in abundance still holds regardless of the location/exposure levels, suggesting that this OTU could be used as a general biomarker of FT tolerance. In fact, the statistical hierarchy of discussing interaction and main effects that the reviewer correctly pointed out must be respected. However, when the direction of the effect is the same between “nested” effects (i.e. location), the main effect suggests a more general role of the OTU.

Ln53: Ergovaline is important and receives more attention but other alkaloids may be just as important in FT. I do not think we have any work that clearly shows that is just ergovaline that is responsible for FT.

AU: We agree. As suggested, we included a sentence in the introduction stating that other ergot alkaloids may be important for FT as well (L522-525 in the revised manuscript) .

Ln81-82: change to: Studies published on the effect of toxic tall fescue on gastrointestinal (GI) microbiota are still limited

AU: Changed as suggested.

.

Ln86-86: not “alleviate some of the impact of FT symptoms” but …reduce the toxic effects of the alkaloids and consequent reduce FT…. or something like that.

AU: Changed as suggested.

Ln87-88: what about the rumen protozoa population? Can they have any impact?

AU: We agree that protozoa might have an impact on FT as well. In this study we focused on bacteria and fungi, and thus didn’t mention the protozoa as our primers for ITS-1 are targeting mostly fungi (and not protozoa).

Ln89: If the microbes capable of degrading the alkaloids are in the rumen, why are you focussing on the feces and not on the rumen microbial populations?

AU: The reviewer is correct that studying rumen microbiota would be of highest relevance for a better understanding of FT. However, as outlined in the materials and methods and discussion sections of our manuscript, we were not able to sample rumen content as the number of animals (n=149) was too high. We would like to refer the reviewer to the respective sections in the revised manuscript for details (L506-514 in the revised manuscript). We also state the limitations of using fecal microbiota for studies like this.

Ln108-110: suggest changing it to: “Cattle were managed in a rotational grazing system and were moved to a new paddock every two weeks at each location to ensure adequate forage management as well as sufficient forage availability to the cows.”

AU: Changed as suggested.

Line118: …as described by Rottinghaus et al [22].

AU: Changed as suggested.

Ln129-138: I wonder how days of gestation would affect those values. What was the variability in gestation days of those cows? Why wasn’t body condition score considered in the model? AWG may not be the best way to assess performance or resistant to FT of those cows. If they were growing steers or heifers yes.

AU: Thanks for the comments. When performing the statistical analysis for the identification of extreme performance cows, we used the effects of parity (which is highly correlated with age) and initial body weight (which is highly correlated with stage of gestation after controlling for parity). Nonetheless, we had evaluated different ways of identifying the contrasting animals for this study. One of them was including the stage of gestation in the model. However, since we were already adjusting for the effects of parity and initial body weight, the effect of stage of gestation did not play any role in redefining the selected individuals. Although we did not formally test BCS in the model, the rationale is similar (BCS is correlated with body weight, which is also impacted by the age/parity). Unfortunately, given that animals were selected for subsequent microbiome analysis (and other analyses not related to this manuscript). We added in the text some explanation why we used parity and initial body weight in the model to try to adjust the data in order to identify animals with contrasting performance (L138-140 in the revised manuscript). 

With regards to the use of AWG of mature cows for selection, we agree that this might not be the most accurate way to meet with our goals. Given the logistics of the research trial at the time, we needed to make a timely decision. Ideally, we wanted to measure growth in their progeny to then retro-actively identify the cows with contrasting performance. This, however, was not possible. Nonetheless, Mayberry (2018; https://repository.lib.ncsu.edu/handle/1840.20/35773?show=full), using the performance data from this project, showed how cows selected as tolerant (hence, greater AWG) had calves with greater growth. Interestingly, this difference existed only in the location where the levels of toxicity were high, whereas this classification had no effect where the levels of toxicity were low. However, due to problems with Mr. Mayberry leaving for a job, in addition of others personal problems, his research paper showing this has not yet been submitted for publications showing these results. In addition, it is worth noting that there are not studies pre-defining how tolerance to fescue should be measured. Hence, we are proposing a methodology in order to advance our knowledge in this very relevant subject for the US beef industry.

Ln256: Change to: All statistical analyses were performed using SAS 9.4 (SAS Institute Inc., Cary, NC).

AU: Changed as suggested.

Ln306-307: you can say there was a decreased species richness (Chao, P=0.0078; ACE, P=0.0093) in the LT cattle for both sites. Because there was as treatment x location interaction. When that happens, you have to report the result and discuss the interaction and not the individual treatment effect.

AU: Addressed in our response to the general comment from the reviewer above.

Ln310-312: The same thing here. Need to focus on the interaction when that is significant.

AU: Addressed in our response to the general comment from the reviewer above.

387-388: …between groups of HT and LT cattle were observed for…

AU: Changed as suggested.

Ln396-402: I suggest changing the description of OUT numbers throughout the whole manuscript to their classification. The reader does not know and need to know what is OTU 1 or 2 or 3 and so on. Change this to the classification of each OUT as described in you materials and methods. Again, if there is an interaction you cannot report results of the main treatment alone. This needs to be re-written in the whole manuscript as the way is written it is wrong and confusing.

AU: We disagree with the reviewer’s comment regarding the OTU numbers: We specifically refer to specific OTUs with their numbers as there can be multiple OTUs within a given genus or family and the OTUs provide the highest taxonomic resolution in our dataset. We believe that by removing the OTU numbers, we will lose resolution and specificity of our results as – as mentioned before – the OTUs are the units with highest taxonomic resolution and summarizing them might blur important biological differences as even closely related phylotypes can have substantially different functions.

Interaction was addressed in our response to the general comment from the reviewer above.

Ln:495: delete “e.g.”

AU: Changed as suggested.

Ln587-593: can it be that Epichloë coenophiala was just digested by the ruminal microbiota? Why would you expect to find it in feces? Lots of things are happening before it reaches the feces.

AU: We agree, we cannot exclude that Epichloe might have been degraded by the rumen microbiota. We have included this possibility explaining the absence of Epichloe sequences in the revised version of the manuscript.

---

## [Decision Letter · Decision Letter 1]

24 Jun 2020

PONE-D-20-02787R1

Beef cattle that respond differently to fescue toxicosis have distinct gastrointestinal tract microbiota

PLOS ONE

Dear Dr. Schmitz-Esser,

Thank you for submitting your manuscript to PLOS ONE. After careful consideration, we feel that it has merit but does not fully meet PLOS ONE’s publication criteria as it currently stands. Therefore, we invite you to submit a revised version of the manuscript that addresses the points raised during the review process.

We look forward to receiving your revised manuscript.

Kind regards,

Marcio de Souza Duarte

Academic Editor

PLOS ONE

Reviewers' comments:

Reviewer's Responses to Questions

**Comments to the Author**

1. If the authors have adequately addressed your comments raised in a previous round of review and you feel that this manuscript is now acceptable for publication, you may indicate that here to bypass the “Comments to the Author” section, enter your conflict of interest statement in the “Confidential to Editor” section, and submit your "Accept" recommendation.

Reviewer #2: All comments have been addressed

Reviewer #3: All comments have been addressed

2. Is the manuscript technically sound, and do the data support the conclusions?

Reviewer #2: Yes

Reviewer #3: Yes

3. Has the statistical analysis been performed appropriately and rigorously? 

Reviewer #2: Yes

Reviewer #3: Yes

4. Have the authors made all data underlying the findings in their manuscript fully available?

Reviewer #2: Yes

Reviewer #3: Yes

5. Is the manuscript presented in an intelligible fashion and written in standard English?

Reviewer #2: Yes

Reviewer #3: Yes

6. Review Comments to the Author

Reviewer #2: Ln63-64: Re-word it. It is a bit confusing.

Ln259: re-write. Confusing. It reads like there was a significant interaction of T and the main effect of T.

Ln260-266: This may just be a result of the ruminal adaptation of animals to the diet. It is questionable if using the w1-7 is the best option in the long term.

Ln318: “However, . we”

Results section. A T*L interaction should not make it difficult to interpret. As there are differences in the Alkaloids in total concentration and composition between sites, which could ultimately be promoting those differences. For example, for the Chao and ACE the interaction you have shows that there is no difference between HT and LT for the BBCFL location but there was a difference between HT and LT groups for the UPRS location. This interaction result description needs to be improved.

Ln482-484: where? Rumen, fecal??

L497-498: it is not difficult. You have huge differences in alkaloids concentration and composition between locations. You need to discuss how that is affecting your results.

Reviewer #3: ABSTRACT

Pag 2, ,lines 31-33: “20 HT and 20 LT cattle balanced by farm were selected fo 16S rRNA gene and ITS1 region Illumina MiSeq amplicon sequencing to compare the fecal microbiota of the two tolerance groups.”

This mind of information is quite omitted in other studies in this field of research. Here, the authors are concerned with simply and objectively describing the procedures from rRNA analysis.

Pag 2, lines 38-39: “This study also found more pronounced shifts in the microbiota in animals receiving higher amounts of the toxin.”

It is a relevant question, since the authors reported all bioethics procedures used under this kind of experiment.

Pag 2, lines 41-42: “Our results thus suggest that some fungal phylotypes might be involved in mitigating fescue toxicosis.”

Maybe the aims of this manuscript would be better described in order to be compared with reported conclusion.

INTRODUCTION

Pag 3, lines 52-53: “However, until now, there is no clear evidence that ergovaline is the most or only responsible ergot alkaloid inducing FT – other ergot alkaloids may also contribute to FT.”

Maybe these issues would be better highlighted and associated with the tested hypothesis of the present manuscript. However, the complement of this text is high explicative and supplies the mentioned comment.

Pag 3, lines 61-63: “Additionally, researchers focused on the endophyte, identifying strains that produce lower levels of the ergot alkaloids while still providing drought and insect resistance for the grass”

I suggest including relevant references if the same are suitable to be exploited over here.

Pag 4-5, lines 89-91: “Our goal was to identify shifts in bacterial, archaeal, and fungal microbial populations 90 (using 16S rRNA gene and ITS1 region amplicon sequencing, respectively) between the two tolerance groups across two different locations.”

The term “Our goal was” would be better exploited in the ABSTRACT section.

MATERIALS AND METHODS

Line 129: If the residual term is presented under an algebraic way, i.e. eijk, the correct notation is the following: eijk ~ N (0, Sig2e). In fact, there is no reason to applied matrix notation over here “where I is the identity matrix. Statistical analysis was performed” (such as at line 129)

Pags 6-7, lines 135-137: “The aim of this study was to compare 136 the fecal microbiota of those animals that showed most extremes in their 137 performance (based on AWG), to achieve a clearer biological signal.”

This kind of information is not “highlighted” as ABSTRACT section. Additionally, see comments at Pag 4-5, lines 89-91. There is some kind of redundancy in the general aims of the present manuscript.

Line 147: The residual term distribution was not reported here.

Line 156: The residual term distribution was not reported here.

RESULTS/DISCUSSION

Both section are very well written and take into account the “essence” of this study. All information exploited in these sections are clear and very well evidenced for the readers.

7. PLOS authors have the option to publish the peer review history of their article (what does this mean?). If published, this will include your full peer review and any attached files.

Reviewer #2: No

Reviewer #3: **Yes: **Fabyano Fonseca e Silva

---

## [Author Response · Author response to Decision Letter 1]

29 Jun 2020

Reviewer #2: 

Ln63-64: Re-word it. It is a bit confusing.

Response: As suggested, we have reworded this sentence and hope it is clearer now.

Ln259: re-write. Confusing. It reads like there was a significant interaction of T and the main effect of T.

Response: Changed as suggested.

Ln260-266: This may just be a result of the ruminal adaptation of animals to the diet. It is questionable if using the w1-7 is the best option in the long term.

Response: It could be that the underlying biological process that resulted in large variability in response to FT is indeed the ability of each animal adapting to the diet. This, therefore, could actually have a genetic basis. However, this area of research is still limited, and we hope our results can help others building more knowledge blocks on top of it so one day we can understand the reasons why there is large individual variability in response to FT. With regards to using 1_7, our decision is based on the extensive literature about variation to stressors, where it is well known that the period of larger variability indicates the most extreme stress period.

Ln318: “However, . we”

Response: changed as suggested.

Results section. A T*L interaction should not make it difficult to interpret. As there are differences in the Alkaloids in total concentration and composition between sites, which could ultimately be promoting those differences. For example, for the Chao and ACE the interaction you have shows that there is no difference between HT and LT for the BBCFL location but there was a difference between HT and LT groups for the UPRS location. This interaction result description needs to be improved.

Response: As suggested by both reviewers in the first round of revisions of this manuscript, we have changed this section to give more priority on reporting the interaction effects over the T or L effects alone. We have thus modified this section accordingly and included more general discussion regarding possible reasons for the differences in diversity and abundance in the discussion (L539-548). However, as we wanted to be cautious and try not to “overinterpret” our results, we have changed this section to accommodate the reviewer’s comment, but also worded this cautiously as we cannot easily discern which effect is driving the T*L interaction effect differences observed. 

Ln482-484: where? Rumen, fecal??

Response: In fecal samples – changed accordingly.

L497-498: it is not difficult. You have huge differences in alkaloids concentration and composition between locations. You need to discuss how that is affecting your results.

Response: Thank you for the comment. Although the differences in toxin levels at each farm are largely different, the difference in abundance noted in this OTU may also be caused by different management practices or climate at the different locations, as OTU15 had opposing effects for location. Without a significant main effect or similar patterns seen in the location to explain the interaction, we feel it may be misleading to readers to explain this difference based on toxin levels alone. We thus prefer to be not too speculative about possible reasons for the differences in abundance of this OTU.

Reviewer #3: 

ABSTRACT

Pag 2, ,lines 31-33: “20 HT and 20 LT cattle balanced by farm were selected fo 16S rRNA gene and ITS1 region Illumina MiSeq amplicon sequencing to compare the fecal microbiota of the two tolerance groups.”

This mind of information is quite omitted in other studies in this field of research. Here, the authors are concerned with simply and objectively describing the procedures from rRNA analysis.

Response: We changed this sentence to more simply and objectively describe the approach in the abstract.

Pag 2, lines 38-39: “This study also found more pronounced shifts in the microbiota in animals receiving higher amounts of the toxin.”

It is a relevant question, since the authors reported all bioethics procedures used under this kind of experiment.

Response: Thank you for your comment.

Pag 2, lines 41-42: “Our results thus suggest that some fungal phylotypes might be involved in mitigating fescue toxicosis.”

Maybe the aims of this manuscript would be better described in order to be compared with reported conclusion.

Response: As suggested, we have included a sentence in the abstract of the revised manuscript highlighting the aims of the study (L28 in the revised manuscript).

INTRODUCTION

Pag 3, lines 52-53: “However, until now, there is no clear evidence that ergovaline is the most or only responsible ergot alkaloid inducing FT – other ergot alkaloids may also contribute to FT.”

Maybe these issues would be better highlighted and associated with the tested hypothesis of the present manuscript. However, the complement of this text is high explicative and supplies the mentioned comment.

Response: Thank you for your comment. As suggested by the reviewer above, we have clarified the aim of the study in the abstract. We hope new research can continue helping all of us better understanding all factors associated with FT. 

Pag 3, lines 61-63: “Additionally, researchers focused on the endophyte, identifying strains that produce lower levels of the ergot alkaloids while still providing drought and insect resistance for the grass”

I suggest including relevant references if the same are suitable to be exploited over here.

Response: As suggested we have included three relevant references in the revised manuscript.

Pag 4-5, lines 89-91: “Our goal was to identify shifts in bacterial, archaeal, and fungal microbial populations 90 (using 16S rRNA gene and ITS1 region amplicon sequencing, respectively) between the two tolerance groups across two different locations.”

The term “Our goal was” would be better exploited in the ABSTRACT section.

Response: Similar to what the reviewer suggested before, we have now included the aim of this study in the abstract.

MATERIALS AND METHODS

Line 129: If the residual term is presented under an algebraic way, i.e. eijk, the correct notation is the following: eijk ~ N (0, Sig2e). In fact, there is no reason to applied matrix notation over here “where I is the identity matrix. Statistical analysis was performed” (such as at line 129)

Response: As suggested, we have reworded this sentence and removed the reference to the identity matrix.

Pags 6-7, lines 135-137: “The aim of this study was to compare 136 the fecal microbiota of those animals that showed most extremes in their 137 performance (based on AWG), to achieve a clearer biological signal.”

This kind of information is not “highlighted” as ABSTRACT section. Additionally, see comments at Pag 4-5, lines 89-91. There is some kind of redundancy in the general aims of the present manuscript.

Response: As suggested by the reviewer in comments before, we have reworded the abstract to include the aims of this study. We have also reworded the sentence the reviewer refers to in this comment to make the reason why we are focusing on the extreme performers and their fecal microbiota.

Line 147: The residual term distribution was not reported here.

Response: As suggested, we have included the residual term distribution.

Line 156: The residual term distribution was not reported here.

Response: As suggested, we have included the residual term distribution.

RESULTS/DISCUSSION

Both section are very well written and take into account the “essence” of this study. All information exploited in these sections are clear and very well evidenced for the readers.

Response: Thank you for your comment.

---

## [Editor Report · Decision Letter 2]

8 Jul 2020

Beef cattle that respond differently to fescue toxicosis have distinct gastrointestinal tract microbiota

PONE-D-20-02787R2

Dear Dr. Schmitz-Esser,

We’re pleased to inform you that your manuscript has been judged scientifically suitable for publication and will be formally accepted for publication once it meets all outstanding technical requirements.

Kind regards,

Marcio de Souza Duarte

Academic Editor

PLOS ONE

Additional Editor Comments (optional):

All minor changes were addressed accordingly. The manuscript is ready to be published.
---

## [Editor Report · Acceptance letter]

10 Jul 2020

PONE-D-20-02787R2 

Beef cattle that respond differently to fescue toxicosis have distinct gastrointestinal tract microbiota 

Dear Dr. Schmitz-Esser:

I'm pleased to inform you that your manuscript has been deemed suitable for publication in PLOS ONE. Congratulations! Your manuscript is now with our production department. 

Kind regards, 

on behalf of

Dr. Marcio de Souza Duarte 

Academic Editor

PLOS ONE